# Epigenetic Regulation by Polycomb Complexes from Drosophila to Human and Its Relation to Communicable Disease Pathogenesis

**DOI:** 10.3390/ijms232012285

**Published:** 2022-10-14

**Authors:** Aaron Scholl, Sandip De

**Affiliations:** Division of Cellular and Gene Therapies, Center for Biologics Evaluation and Research, Food and Drug Administration, Silver Spring, MD 20993, USA

**Keywords:** Polycomb, PcG, communicable, pathogenesis

## Abstract

Although all cells in the human body are made of the same DNA, these cells undergo differentiation and behave differently during development, through integration of external and internal stimuli via ‘specific mechanisms.’ Epigenetics is one such mechanism that comprises DNA/RNA, histone modifications, and non-coding RNAs that regulate transcription without changing the genetic code. The discovery of the first Polycomb mutant phenotype in Drosophila started the study of epigenetics more than 80 years ago. Since then, a considerable number of Polycomb Group (PcG) genes in Drosophila have been discovered to be preserved in mammals, including humans. PcG proteins exert their influence through gene repression by acting in complexes, modifying histones, and compacting the chromatin within the nucleus. In this article, we discuss how our knowledge of the PcG repression mechanism in Drosophila translates to human communicable disease research.

## 1. Introduction

Epigenetics, the chemical modification of DNA/RNA and its associated proteins, is one of the key mechanisms that regulates spatiotemporal expression of genes in an organism. One of the most studied epigenetic modifications is tri-methylated lysine 27 of histone 3 (H3K27me3). This mark is deposited in Drosophila by Polycomb group (PcG) protein complexes. The first Polycomb phenotype, extra sex comb (esc), which is associated with a recessive mutation, was discovered in male flies which had partial sex combs on the second and third pairs of legs [1]. After a few years, a similar phenotype associated with a dominant mutation was discovered and named Polycomb (Pc) [2]. The conceptual breakthrough in the field came in the year 1978, when it was proposed that Pc encodes a repressor of the bithorax complex gene in Drosophila [3]. Subsequent genetic and molecular investigations have shown that PcGs are a group of proteins that are conserved from Drosophila to humans. PcGs control developmental pathways, enhance cellular proliferation and stem cell maintenance, inhibit apoptosis, and contribute to the regulation of other diverse cellular functions. PcG proteins and cancer are strongly associated; various cancer types show dysregulation of PcG protein expression and function [4,5,6,7,8].

In this review, we will primarily focus on the comparison between human and Drosophila Polycomb-mediated gene silencing mechanisms, as well as how Drosophila has contributed to our understanding of the human PcG mechanism. Although a great deal of research and reviews has been published on the role of Polycomb complexes in human cancer using Drosophila as a model, little has been written about the pathophysiology of communicable diseases and how Polycomb-mediated gene repression relates to it. In this article, we will introduce several infectious diseases and discuss how they are reported to affect the PcG mechanism.

## 2. Polycomb Protein Complexes

PcG proteins assemble into two types of Polycomb Repressive Complex (PRC), which are highly modular and have mutually reinforcing interactions. PRC1 is an E3 ubiquitin ligase that serves to mono-ubiquitinate histone H2A in Drosophila and humans at lysine 118 and 119, respectively [9,10,11]. PRC2 acts as a methyltransferase that variably mono-, di-, or tri-methylates histone H3 at lysine 27 (H3K27me1/2/3) [12,13,14,15]. The cores of both complexes exhibit a high degree of homology between Drosophila and mammals.

Drosophila and human canonical PRC1, respectively, exhibit structural and functional homology between Polycomb (Pc) and Chromobox (CBX), Sex combs extra (Sce or dRING) and Really Interesting New Gene (RING1A/B), Posterior sex combs or Suppressor of zeste 2 (Psc)/Su(z)2, and Polycomb Group Ring Finger (PCGF2/4), and Polyhomeotic (Ph) with Polyhomeotic-like protein (PHC (1–3)) groups. Pc and CBX groups interact with the conserved RING complex and recognize H3K27me3 [16]. Sce and RING1A/1B monoubiquitinate H2A (H2AK118Ub1 and H2AK119Ub1 in Drosophila and humans, respectively) [17]. PSC/Su(z)2 and PCGF share similar functions in maintaining the overall PRC1 complex [18]. Ph and PHC (1–3) may contribute to higher-order chromatin structure formation, leading to transcriptional repression [19].

Mammalian PRC1 is a highly variable protein complex with canonical and noncanonical variants. The canonical PRC1 complex in mammals contains a variable Chromobox protein (CBX2, CBX4, CBX6, CBX7, CBX8), a polyhomeotic-like protein (PHC1, PHC2, PHC3), a Polycomb group RING finger (PCGF2, PCGF4), and an E3 ubiquitin ligase RING group (RING1A, RING1B) (Figure 1). Canonical variants of PRC1 contain only one of the five potential CBX protein subunits which function in recognizing PRC2 target loci (H3K27me3) [12,13,15,16,20]. Each CBX protein subunit of PRC1 has expression patterns that are more prominently expressed in certain cell types [21,22]. Noncanonical variants of PRC1 lack CBX proteins, but instead contain a RING1 and YY1 Binding Protein (RYBP) or YY1 associated factor 2 (YAF2), which compete for the same RING1A/B interaction site as CBX proteins (Figure 1) [23]. Polycomb group RING fingers PCGF2 and PCGF4 stabilize the PRC1 complex [24]. Canonical and noncanonical variants of PRC1 differ in their dependency on H3K27me3 marks deposited by PRC2. Noncanonical PRC1 variants also differ in the PCGF (1–6) present within the complex. These variations in each of the components of the PRC1 complex alter the function and targets of each complex.

PRC2 is similarly modular, with a high degree of homology existing between the Drosophila and mammalian complexes. PRC2 functions to variably mono-, di-, and tri-methylate lysine 27 on histone H3 (H3K27me1–3). The core of the mammalian PRC2 complex consists of EED (1–4), EZH1/2, SUZ12, and RBAP46/48. EED (1–4) are the mammalian counterparts of Drosophila Extra sex combs (Esc). EZH1/2 are the mammalian homologs of Enhancer of zeste (E(z)) (Figure 1) [25]. Mammalian SUZ12, equivalent to Drosophila Su(z)12, is essential for guiding PRC2 to its correct genomic loci (Figure 1) [26]. Retinoblastoma-associated protein 46 and 48 (RBAP46/48) are comparable to Nurf55 in Drosophila, and, in the context of PRC2, interact with the histone H4 tail [27]. Various PRC2-associated proteins such as JARID2, AEBP2, and PCL (1–3) share homology within the Drosophila system as well. Table 1 represents all the Polycomb complex proteins, and their abbreviated names in both human and Drosophila, with their activities.

## 3. Recruitment of Polycomb Complexes to Specific Sites in the Genome

In Drosophila, PcG proteins are recruited to target genes via specific DNA fragments known as Polycomb Response Elements (PREs) (Brown, J.L., Kassis, J.A. 2013). PREs contain consensus motifs for multiple DNA binding proteins (Pho, Phol, Trl/GAF, Spps, Dsp1, and Cg) that are known to recruit PRC2, which in turn deposit H3K27me3, eventually recruiting PRC1 through its chromodomain-containing CBX homolog, PC (Polycomb) [10]. Afterwards, Sce, the ortholog of RING1A/B, mono-ubiquitinates H2AK118 (in mammals, H2AK119). This hierarchical recruitment model in Drosophila [10], with inputs from genome-wide studies, is eventually converted to a context-dependent cooperative model, where a dynamic relationship between PREs, DNA binding proteins, PRC complexes, and genomic context help the recruitment and stabilization of PRCs to their target sites [49,50].

In mammals, reports on PREs are rather limited [51,52,53]. The evidence currently available indicates that hypomethylated CGIs function as mammalian PREs [54] and interact with vPRC1.1 through the zinc finger protein KDM2B [55,56]. Chromatin recruitment of the vPRC1.6 complex requires the coordinated actions of its MGA-MAX and E2F6-DP1 subunits, and recruitment of the vPRC1.3 complex containing PCGF3 requires an interaction with the USF1 DNA binding transcription factor [57,58,59]. Canonical PRC1s are recruited to the target sites through CBX proteins binding to H3K27me3 [60]. PCL proteins PCL1 (also known as PHF1), PCL2 (also known as MTF2), or PCL3 (also known as PHF19) preferentially bind CGIs in order to promote PRC2 binding and to stabilize the dimerization of PRC2 [61,62,63]. Further, SUZ12 and JARID2 of PRC2 also have an affinity for GC-rich DNA sequences [26,64]. In mammals, the PRC1-dependent H2AUb mark is recognized by Jarid2 and AEBP2 of PRC2. This crosstalk reinforces the recruitment of these complexes to their target sites [65,66,67]. Although controversial, both short and long non-coding RNAs are also reported to regulate PRC recruitment in mammals [68,69]. The most studied example of RNA-mediated PRC recruitment comes from the X chromosome inactivation in mammals, where long non-coding RNA Xist drives recruitment of PRC complexes [69,70,71]. In parallel, alternative mechanisms are proposed to drive the binding of PRCs to the CGIs. One such mechanism is ‘chromatin sampling,’ which proposes that Polycomb complexes recruit at permissive chromatin locations that lack antagonistic chromatin modifying activities associated with active transcription [72].

## 4. Mechanism of Polycomb Domain Establishment and Target Gene Repression

PcG complexes use a variety of methods to inhibit transcription once they have been recruited to their target loci. One of the most studied and referenced mechanisms is transcriptional repression via the establishment of Polycomb domains through chromatin compaction. Following chromatin compaction, the chromatin remodeling and access of the underlying genes to transcription activation factors are restricted, which drives transcriptional silencing. So, how do PcG complexes drive chromatin compaction? After recruitment of PRC2 complexes, E(z)/EZH2 modifies H3K27 with a trimethylation mark. The Eed/Esc subunit of PRC2 binds to the deposited H3K27me3 mark through its aromatic cage and allosterically stimulates binding of the PRC2 [73,74]. This positive feed-forward mechanism progresses bi-directionally from the PREs/CGIs, spreading the mark locally along the chromatin fiber [75,76] (Figure 2). Once deposited, H3K27me3 is maintained through DNA replication by PRC2-dependent replenishment of the mark on newly incorporated unmarked histones [77]. As a result, H3K27me3 survives the cell divisions and provides transgenerational epigenetic memory [78]. At least in Drosophila, the mere presence of H3K27me3 is not sufficient to indefinitely propagate the Polycomb mark, and thus requires PcG-recruiting DNA fragments to maintain the chromatin memory through cell divisions [79]. In flies, a point mutation in lysine 27 of H3 fails to repress PRC2-target genes, mimicking PRC2 mutant phenotypes and indicating that H3K27 is required for PcG repression [80].

H3K27me3 also acts as the anchoring site for chromodomains present in Pc/CBX of PRC1/cPRC1, which in turn deposits the H2AUb mark over its target region. Although H2Aub is not required in PRC1 target genes repression [81,82], it is well documented that the core components of the PRC1 complex induce compaction of defined nucleosomal arrays [83]. CBX2 of the cPRC1 contains a highly positively charged nucleosome compaction region, and mutation in this region causes defects in PcG-mediated repression [84]. In addition, Ph/PHC (1–3) proteins in the cPRC1 contain sterile α-motif domains, which interact with other cPRC1 complexes and are proposed to drive the establishment of higher-order chromatin structure through establishing chromatin loops genome-wide [85,86,87,88,89].

In both Drosophila and mammals, the PcG target regions contain several PcG recruitment sites (strong and weak PREs in Drosophila, as well as strong and weak nucleation sites in mammals). Chromosome-conformation capture studies have shown that these sites spatially interact with each other [90,91]. The current hypothesis is that, during development, interaction between these PcG recruitment sites through PRCs in near-cis and far-cis, as well as simultaneous spreading of the H3K27me3 mark, drive the establishment of Polycomb domains, chromatin compaction, and gene silencing [90,92] (Figure 2). Further evidence suggests that L3MBTL2, a subunit of the vPRC1.6 complex, can mediate chromatin compaction independently of histone post-translational modifications [93]. PcG complexes use other strategies for transcription repression. For example, Polycomb inhibits histone acetylation by CBP by binding directly to its catalytic domain [94]. PcG proteins are also proposed to block RNA PolII initiation [95,96,97].

The topic of ‘Polycomb Group protein complexes and its role in gene repression and genome organization’ has been covered in much more detail in several recent reviews [4,5,28,98,99,100].

## 5. Role of Histones in Pathogenic Infection

Pathogenic infection occurs when a bacteria or virus invades a host organism. The ability of a pathogen to target and hijack epigenetic control from the host is paramount to the long-term survival of the pathogen. As viruses proliferate and adapt, host epigenetic mechanisms become critical targets for pathogens to modify for optimal survivability during the early and late stages of infection. There are many mechanisms through which pathogens interact with histones and other host factors during acute and persistent infections. For example, toxins from Listeria monocytogenes, Clostridium perfringens, and Streptococcus pneumoniae dephosphorylate histone H3 (H3S10) and deacetylate histone H4 during early phases of infection, facilitating further infection [101]. Activation of phosphatase PP1 by dephosphorylating threonine 320 (T320) and the subsequent H3S10 dephosphorylation are presumed to be a general epigenomic mechanism of intracellular survival among pathogenic bacteria [102]. Persistent viral infections, such as herpes simplex viruses, interact with the host through many mechanisms, including through PcG members, to epigenetically alter certain host cell types, facilitating the viral replication, latency, and eventually reactivation of the virus [103]. The Epstein–Barr virus EBNA3A inactivates intergenic enhancers in human B cells, which initiate and maintain multiple direct or indirect PcG signatures across a chromatin domain that includes chemokine genes CXCL10 and CXCL9 [104]. This transcriptional repression of chemokine genes prevents the attraction of white blood cells to sites of infection. Pathogenic epigenetic alterations have the potential to alter the reprogramming potential of B cells, further impacting immune response to pathogenic infection [105]. As histone modification is a viable method of epigenetic control by pathogens, it is not surprising that PcG members are also pathogenic targets.

## 6. Role of PcGs in Pathogenic Infection

Genes suppressed by PcG complexes can transition between the active and repressed states of transcription in different cell and tissue types and at different times, a process known as ‘facultative heterochromatinization’. During their infection cycles, some human viruses can transition between the lytic and lysogenic stages. Thanks to the host PcG mechanism, these viruses have a special opportunity to make use of this system for their own purposes. Because of this, the link between a few specific virus types and host PcGs has been extensively explored.

### 6.1. HIV Latency and Polycomb

Human immunodeficiency virus (HIV) is a retro virus that attacks the immune system in the human body, weakening a person’s immunity against opportunistic infections. Numerous studies carried out over the last two decades have demonstrated that HIV remains integrated and replication competent in a small number of host cells, and this is regarded as the latent reservoir of HIV infection [106]. PcG proteins have recently received much attention due to their involvement in the regulation of HIV-1 latency [107]. It has been shown that H3K27me3 and EZH2 are required for the maintenance of HIV-1 latency [39,40]. In latently infected cells, disruption of PcG silencing reactivates the HIV-1 Provirus [40,108,109]. The human PcG protein EED interacts with the integrase of HIV type 1, enhances the HIV-1 integration process [37], and negatively affects HIV-1 assembly and release [38]. Furthermore, a possible linkage between the HIV-1 latency and increased expression of PRC1 proteins BMI1 and RING2, as well as associated upregulation of ubiquitylation at histone H2A, have been shown [110]. In addition, several other transcription factors are also linked to HIV latency. For example, CBF-1 promotes the establishment and maintenance of HIV latency by recruiting PRC1 and PRC2, at HIV long-terminal repeats [111]. YY1 and LSF interact with each other and form a complex known as repressive complex sequences (RCS). RCS specifically and synergistically represses HIV-1 from the long terminal repeat via recruitment of histone deacetylase 1 [112,113]. Due to its involvement, PcGs have become an attractive and promising therapeutic drug target in the eradication of HIV latent reservoirs [114,115]. An overview of pathogen involvement with conserved PRC complexes is shown below in Table 1.

### 6.2. HPV and Polycomb

Human papillomavirus (HPV) is a large group of related DNA viruses. Each virus in this group is given a number; of these, HPV16 and HPV18 are the ‘most high risk’ types, as they are responsible for the greatest number of HPV-related cervical and other cancers in human [116]. HPV16 protein E7 associates with E2F6 and related PcG proteins [117]. Cysteine 91 in the C terminus of HPV16 E7 and the marked box of E2F6 are critical for the functional association of E7 to E2F6, and for the de-repression of E2F6-target genes, by abrogating the repressive activity of the E2F6 [117]. The E2F6 transcription factor is a component of the mammalian Bmi1-containing Polycomb complex [118]. In fact, E6/E7 proteins from HPV16, HPV18, HPV11, and HPV6b also interact with other E2F transcriptional factors and induce EZH2 expression through binding to the EZH2 promoter [41]. This increased expression of EZH2 has been implicated in the progression of HPV-associated cervical cancer [41,42]. In this case, a communicable and persistent viral infection activates a PcG member, leading to the development of cervical cancer and a progressively poorer prognosis due to continuous activation. Interestingly, studies in primary human epithelial cells revealed that E6/E7 expression also drives increased expression of KDM6A (UTX), KDM6B (JMJD3), and EZH2, as well as the reduction in PRC1 protein Bmi1 and total H3K27me3 [31,119]. In addition, H3K27me3 was also lost in high-grade squamous cervical intraepithelial lesions, while EZH2 expression was elevated [31]. These somewhat paradoxical findings were nicely summarized by McLaughlin-Drubin et al. [120]. They hypothesized that HPV16 E6/E7 mediated AKT up-regulation [121] and caused increased phosphorylation at serine (S) 21 of the EZH2, and this in turn suppressed H3K27 methylation activity [122].

### 6.3. Herpes Virus and Polycomb

DNA viruses belonging to the Herpesviridae family infect both humans and other animals. Humans are the main host for eight different Herpes viruses. They are the Herpes Simplex Virus 1 (HSV1), HSV 2, Varicella-Zoster Virus (VZV/HHV3), Epstein–Barr Virus (EBV/HHV4), Cytomegalovirus (HHV5), Human Herpes Virus-6 (HHV6), HHV7, and Kaposi’s Sarcoma Herpes Virus (KSHV/HHV8). Herpes viruses can remain dormant in the host and create recurrent infections [123]. While in latency, Herpes virus genomes associate with cellular histones to create a chromatinized structure. This enables Herpes virus genomes to suppress their lytic genes, avoid being identified as foreign DNA particles, and remain undetected [123,124]. In order to better understand the natural latency of Herpes viruses, the epigenetic structure of viral genomes has been examined in cells and tissues separated from human individuals [43,46,47,125,126]. Epigenetic landscapes of different HSVs have also been investigated in patient-derived cell lines [44,48,127,128]. In addition, small animal models or primary cells from animals were also used to study the epigenetic basis of HSV latency [129,130,131,132,133,134,135].

Conclusively, investigators have found, so far, that at latency, the HSV-1 genome is marked with H3K27me3 and is transcriptionally repressed [129,131,136]. The only gene expressed abundantly during HSV-1 latency is the ‘latency-associated transcript’ (LAT), which produces a long-noncoding transcript LAT [123,124]. While SUZ12 is recruited to the HSV-1 genome during the establishment of latency, the involvement of PRC1 complex proteins in the viral latency establishment needs to be investigated further [131,135]. Additionally, the fact that the H3K27 demethylases, JMJD3 and/or UTX, must be active for HSV-1 to reactivate further supports the idea that H3K27me3 contributes to HSV-1 latency [134,137].

The epigenetic contribution to HSV-2 and HHV-3 latency has not yet been studied extensively [138,139,140,141]. In the case of EBV/HHV4, several studies have detected the presence of H3K27me3 and EZH2 in the lytic promoters of the EBV/HHV4 genomes [127,128]. In addition, the deletion of EZH2 significantly increased viral DNA replication and progeny production, as well as the expression of both latent and lytic viral genes. These findings show that EZH2 is essential for the complex epigenetic regulation of both active and latent gene expression [142]. Although the presence of H2AK119ub1 on the latent EBV genomes has not yet been tested, it is well documented that KDM2B is present at the EBV lytic promoters [35].

Experiments involving disruption of PRC2 activity by chemical methods, genetic knockdown of H3K27 demethylase JMJD3, or overexpression of JMJD3 and UTX suggest that PRC2-mediated repression of viral transcription is a key step in the establishment and maintenance of HCMV/HHV5 latency [43,143,144]. Chromatin immunoprecipitation experiments have also shown that the H3K27me3 mark and the PRC2 proteins EZH2 and SUZ12 are also associated with the major immediate early promoter of the HHV5 latent genome [43]. While deposition of H3K27me3 on the HSV1 and HHV4 genome is well documented, direct evidence regarding accumulation of the same epigenetic mark on the HHV5 genome is comparatively lacking. In case of HHV6, a recent study discovered measurable levels of H3K27me3 on HHV6 DNA in iciHHV-6A (inherited chromosomally integrated HHV6) cells [145,146].

Finally, an extensive body of evidence supporting PcG-mediated silencing during KSHV/HHV8 latency has been gathered by analyzing many model systems [147]. Both H3K27me3 and PRC2 proteins (EZH2 and SUZ12) have clearly been found to be enriched on silenced lytic promoters of the KSHV latent genome [29,44,46,47,48]. Further, EZH2 inhibition, knockdown, or over-expression of the H3K27 demethylases UTX and JMJD3, increase viral gene expression [29,44]. In comparison to other herpes viruses, the connection between KSHV latency and PRC1 and its associated chromatin mark H2AK119ub1 has received significantly greater attention, for unclear reasons. Using chromatin immunoprecipitation, knockdown, and over-expression studies, it has been found that the H2AK119ub1 mark and components of canonical and non-canonical PRC1 complexes RING1B, Bmi1 and KDM2B, RYBP are associated with lytic promoters of the dormant KSHV genome [29,30,32,36,148].

Additional studies have shown that the KSHV long non-coding RNA, PAN RNA, binds to PRC2 through JMJD3 and modulates the host’s inflammation, cell cycle, and immune response to infection [149]. PAN RNA also interacts with UTX and MLL2, which are Trithorax group proteins (TrxGs) [149]. TrxGs are a heterogeneous group of factors with varied activities, mainly related to chromatin modification, associated with active transcription and counteracting repressive roles of PcG proteins [28,99,150,151].

### 6.4. Other Pathogens and Polycomb Regulation

Studying how parasites have developed sophisticated strategies to control host gene transcription and protein expression from an epigenetics perspective is turning out to be an interesting and significant subject of research. A limited number of reports can be found on how other types of pathogens (such as other viruses, bacteria, protozoa, different parasites, etc.) are manipulating PcG-mediated gene repression mechanism. This needs to be explored better in order to fully understand the process of disease pathogenesis, as well as that of using PcG components as drug targets to control the disease. For example, in Ehrlichia chaffeensis, intracellular infection is promoted by the expression of bacterial proteins that recruit PcG proteins for degradation. During early infection, the E. chaffeensis tandem repeat protein 120 (TRP120) interacts with multiple PCGF isoforms, which are components of PRC1, degrading them and destabilizing the PRC1 complexes [33,34]. Experiments targeting EZH2 have shown that the influenza virus genome can be exported from the nucleus with the help of Polycomb Repressive Complex 2 [45]. Interestingly, a recent article has proposed that pharmacologic inhibitors of PRC2, which are currently in advanced clinical trials for cancer treatment, should be tested to treat COVID-19 patients [152]. The rationale for their hypothesis is that both influenza and coronavirus pathogenic strains can activate PRC2 and deposit H3K27me3 at the promoters of certain interferon responsive genes, and repress their transcription [153]. So, anti-PRC2 drugs might help in treating the SARS-CoV2 infection.

As discussed above, PcG members are involved in transcriptional repression and cellular memory, and it is important to identify their roles during pathogenic infection. It is also critical to distinguish the effects of specific PcG members in acute pathogenic infections versus persistent pathogenic infections, as persistent pathogenic infections have the potential for long-lasting epigenetic impacts on human health. In vivo mouse models and cultured mammalian cells have been used in the majority of studies on host–pathogen interactions. This makes sense, given that these systems successfully simulate some elements of the host environment in humans. However, these studies are frequently complex, time consuming, technologically challenging, and expensive. Often, biological pathways with great evolutionary conservation through taxa are impacted by pathogens, which gives us the opportunity to study them in non-mammalian multi-cellular model organisms. Drosophila has been utilized as a model organism to comprehend basic mechanisms of genetics and development due to its short life cycle, low cost, conserved biological pathways, availability of mutant stock, well characterized mutant phenotypes, and accessibility of cutting-edge technologies [154]. In addition, Drosophila has a conserved innate immune system that responds to viral, bacterial, and other pathogens by producing anti-microbial peptides and engaging hemocytes which have phagocytic capacity [155]. Thus, this model organism is also becoming a preferred system for studying human disease pathogenesis [156]. Here, we suggest that the molecular interactions between pathogen effector molecules and various epigenetic pathways be investigated, including Polycomb-mediated gene repression mechanisms, which are highly conserved in Drosophila.

## 7. Conclusions

Realization of the importance of PcGs in organism development, discovery of the links between PcGs and various human cancers, accessibility to new cutting-edge technologies, and, above all, the availability of high capital and human resources has generated an unprecedented amount of interest and opportunity to study the PcG-mediated gene repression mechanism. The identification of new PRC proteins, their ability to form various complexes, their roles in various cell types, and their ability to act cooperatively in a diverse spatio-temporal context has added several layers of complexity to this field. In this review, our attempt was to discuss the various PRC complexes, as well as their role in gene repression and genome organization from Drosophila to humans, in a relatively simple and succinct manner. Furthermore, we aimed to briefly discuss the role of the Polycomb-mediated gene repression mechanism beyond organism development and cancer, in the world of communicable disease. Comprehensive studies focused on understanding the mechanistic contributions of PcG proteins and complexes in various communicable diseases. Exploring the molecular interactions between these complexes and pathogenic effector molecules will be key to understanding their physiological importance, with respect to disease pathogenesis and how they can be therapeutically targeted. We believe that, as Drosophila has contributed immensely to understanding and appreciating almost every aspect of biology, it will also be an asset for studying various infectious diseases and their association with PcG complexes, gene repression, and genome organization driving disease pathogenesis.

## Figures and Tables

**Figure 1 ijms-23-12285-f001:**
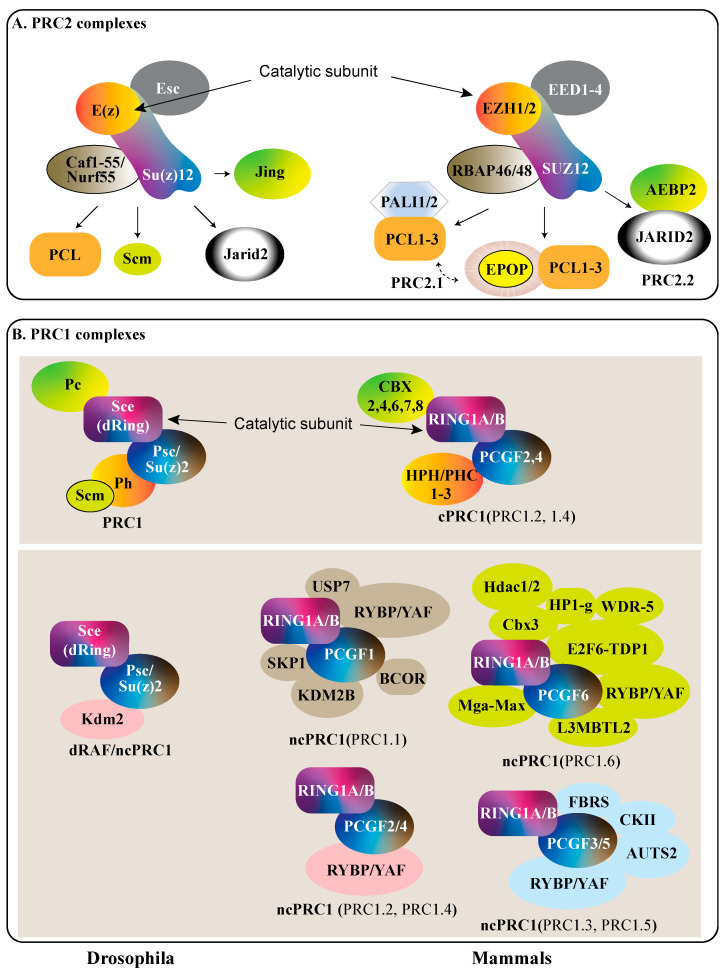
A comparison of Drosophila and mammalian Polycomb complexes. (**A**) PRC2 in Drosophila (left) and mammals (right). The main subunits of PRC2 are conserved between Drosophila and mammals with differences in interacting components. The core-conserved subunits are color coded. (**B**) PRC1 in Drosophila (left) and mammals (right). Canonical (cPRC1) and various forms of non-canonical (ncPRC1) mammalian PRC1 are depicted. Drosophila PRC1 and mammalian cPRC1 exhibit many conserved subunits; conserved subunits are color coded. The various forms of mammalian ncPRC1 share only RING1A/B and one PCGF member in common with cPRC1. Adapted from Kuroda et al. [28].

**Figure 2 ijms-23-12285-f002:**
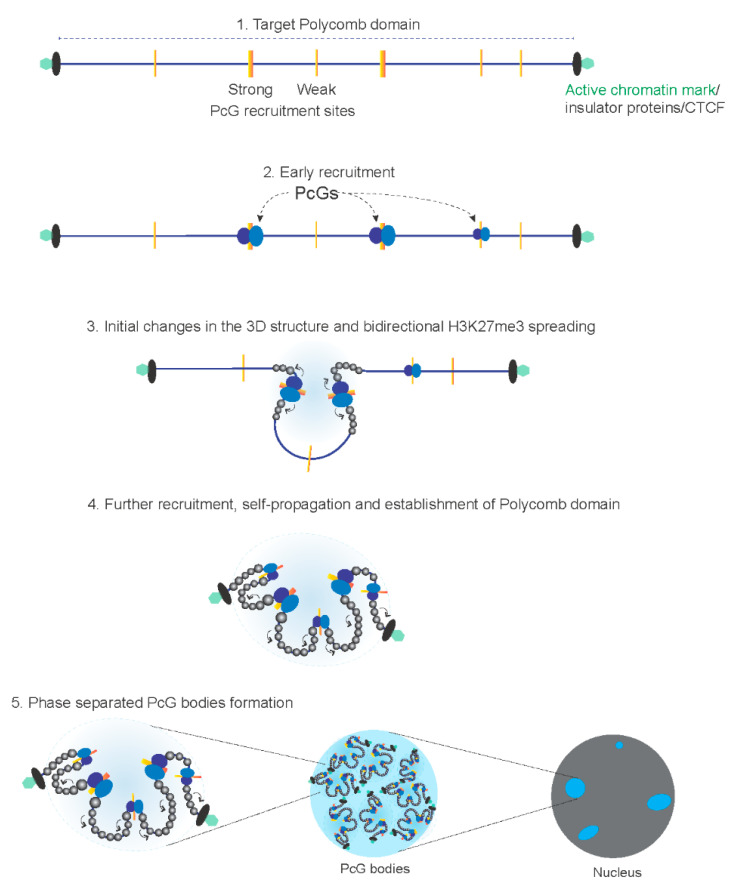
Polycomb domain establishment. (**1**) PcG recruitment sites (strong and weak) within a target Polycomb domain, PcG domain boundaries: in Drosophila, either actively transcribed genes or insulator proteins act as boundaries, and in mammals, CTCF proteins act as boundaries; (**2**) initially, PcGs are recruited to PcG recruitment sites, predominantly to the strong sites; (**3**) 3D structural changes occur due to PcG recruitment, and H3K27me3 spreads bi-directionally from these sites; (**4**) further recruitment leads to H3K27me3 spreading throughout the domain in a feed-forward mechanism, and induces widespread structural changes; (**5**) phase separated PcG bodies are formed throughout the nucleus. Adapted from Kuroda et al. [28].

**Table 1 ijms-23-12285-t001:** PRC subunits, their functions, and pathogen involvement.

*D. melanogaster*	*H. sapiens*	PcG Complex	Function	Pathogen Involvement
Pc	CBX-2,4,6,7,8	cPRC1	Interacts with histone H3K27me3	TBD
Sce	RING1A/B	PRC1	E3 ubiquitin ligase	KSHV [29,30]
Psc	BMI1/PCGF4	PRC1	Stabilize PRC1 complex	HPV [31], KSHV [32], *E. chaffeensis* [33,34]
Su(z)2	MEL18/PCGF2	PRC1	Stabilize PRC1 complex	*E. chaffeensis* [33,34]
Ph-p and Ph-D	HPH/PHC-(1–3)	PRC1	Establish High-order chromatin structure	TBD
Rybp	RYBP	ncPRC1	Interacts with histone modification marks	KSHV [29]
Kdm2	KDM2B/FBXL10	ncPRC1	DemethylatesLysine-36 of Histone H3	EBV [35] KSHV [36]
Esc	EED	PRC2	Binds to H3K27me3	HIV-1 [37,38]
E(z)	EZH1/2	PRC2	Histone methyltransferase of H3K27	HIV-1 [39,40], HPV [41,42], HHV5 [43], KSHV [29,44],Influenza [45]
Su(z)12	SUZ12	PRC2	Stabilizes core complex	HHV5 [43], KSHV [46,47,48]
Caf1-55/Nurf55	RBAP46/48	PRC2	Interact with Histone H4 tail	TBD
Jarid2	JARID2	PRC2	Scaffold for recruitment of complexes	TBD

TBD—to be determined, EBV—Epstein–Barr Virus, HIV-1—Human Immunodeficiency Virus-1, HPV—Human papillomavirus, HHV5—Human Herpes Virus 5, KSHV/HHV8—Kaposi’s Sarcoma Herpes Virus/Human Herpes Virus 8.

## Data Availability

Not applicable.

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
