# Peer review of "Epigenetic Regulation by Polycomb Complexes from Drosophila to Human and Its Relation to Communicable Disease Pathogenesis"

_ijms, 2022, doi:10.3390/ijms232012285_

Round 1

Reviewer 1 Report

In this review the authors aim to present the Polycomb complex: PRC1, and PRC2, their implication in gene silencing and genome organisation in Drosophila and human in a succinct manner. It focuses on how studies on Drosophila contributed to the understanding of polycomb contribution to the pathophysiology of communicable diseases.

After a presentation of the PRC1 and PRC2 complex activities and interactants in both Drosophila and Human, the authors detail the recruitment of Polycomb complexes to specific sites in the genome. They further discuss what is known, or suggested, about polycomb repressive DNA domain establishment, and how they induce chromatin compaction, and/or nucleosomal compaction, leading to gene silencing. They then introduced what has been described about role(s) of histone and PcGs in pathogenic infection. The authors focus their review on the role of PcG in communicable disease infection which have received little attention. The authors present argument supporting Drosophila, which has a conserved innate immune system, as a relevant model system for the study of human disease pathogenesis.

The review is relevant, addresses an important subject poorly covered by the literature on Polycomb proteins. It is well written and structured.  

The review will benefit from a better Figure 1. Specifically, protein activities (Methylation, ubiquitination....) shall be indicated. Moreover considering the large number of PCR complex proteins, and interactants, to complete Figure 1 and to improve clarity of the review it will be useful to have a table with all the Polycomb complex proteins, their names and abbreviations in both human and Drosophila with their activities.

Moreover, adding to the review a recapitulating Table describing infectious diseases that relate to PcG deregulations and the model system where it has been described/studied, will be useful.

Page 6 line 228 229 typo mistakes

Author Response

Dear editor,

Thank you for giving us the opportunity to submit a revised draft of our manuscript titled ‘Epigenetic regulation by Polycomb complexes from Drosophila to human and its relation to communicable disease pathogenesis’ to IJMS.  We appreciate the time and effort that you and the reviewers have dedicated to providing your valuable feedback on our manuscript. We are grateful to the reviewers for their insightful comments on our article. We have been able to incorporate changes to reflect most of the suggestions provided by the reviewers.

Reviewer 1-

  • We have clearly indicated the catalytic subunits of the PRC1 and PRC2 complexes in the Fig. 1.
  • We have added a table to the manuscript with all the Polycomb complex proteins, their abbreviated names in both human and Drosophila with their activities. We have also included a column to the table summarizing PcG subunits and their relation to the pathogens.
  • We have corrected the typo mistake.

Reviewer 2-

  • We have vastly improved the section 6- ‘Role of PcGs in pathogenic infection’. We have carried out an extensive literature search and have expanded on the role of PcGs in pathogenic infection. We hope our new section 6 will satisfy the reviewer.
  • We have added a table to the manuscript with all the Polycomb complex proteins, their abbreviated names in both human and Drosophila with their activities. We have also included a column to the table summarizing PcG subunits and their relation to the pathogens with references.

Sincerely,

Sandip De

Reviewer 2 Report

Scholl and De, in their review entitled “Epigenetic regulation by Polycomb complexes from Drosophila to human and its relation to communicable disease pathogenesis” mentioned they would talk about the translation of knowledge of the PcG repression mechanism in Drosophila to human communicable disease research.

From the title one would expect to see epigenetic regulation by PcG and there should be significant discussion on communicable disease pathogenesis.  

They repeated in abstract, intro, text and conclusion that PcGs in cancer are known and they would discuss several infectious diseases and how they are reported to affect the PcG mechanism. Where was it mentioned? I completely missed it. The only section relevant is ‘6. Role of PcGs in pathogenic infection’. The most important content from line 236 to 246 has no references.  

Most of the review is repetition from previously published reviews.

It cannot be published in any Journal.

Author Response

(The authors gave the same response as above.)

Round 2

Reviewer 2 Report

The manuscript is not significantly improved to be accepted.